# Toward Understanding Most of the Context in Document-Level Neural Machine Translation

**Gyu-Hyeon Choi [1,2,*], Jong-Hun Shin [2], Yo-Han Lee [2] and Young-Kil Kim [2]**

[1] Department of Computer Software, University of Science and Technology (UST), Daejeon 34113, Korea
[2] Electronics and Telecommunication Research Institute (ETRI), Daejeon 34129, Korea; jhshin82@etri.re.kr (J.-H.S.); carep@etri.re.kr (Y.-H.L.); kimyk@etri.re.kr (Y.-K.K.)
[*] Correspondence: choko93@ust.ac.kr

**Abstract:** Considerable research has been conducted to obtain translations that reflect contextual information in documents and simultaneous interpretations. Most of the existing studies use concatenation data which merge previous and current sentences for training translation models. Although this corpus improves the performance of the model, ignoring the contextual correlation between the sentences can disturb translation performance. In this study, we introduce a simple and effective method to capture the contextual correlation of the sentence at the document level of the current sentence, thereby learning an effective contextual representation. In addition, the proposed model structure is applied to a separate residual connection network to minimize the loss of the beneficial influence of incorporating the context. The experimental results show that our methods improve the translation performance in comparison with the state-of-the-art baseline of the Transformer in various translation tasks and two benchmark machine translation tasks.

**Keywords:** BERT; cardinality residual connection; context-aware machine translation; document-level neural machine translation; sentence embedding; similarity measurement; Transformer





## 1. Introduction

The neural machine translation (NMT) model [1,2] has been extended to obtain translated texts that reflect contextual information. Recently, the development of language models learned through abundant language resources has improved the ability to recognize contexts. Pretrained language models have shown superior performance and have attracted attention for use in various natural language processing tasks. Most existing models [3–7] for document-level machine translation use two encoders to model source sentences and document-level context. The standard Transformer model is extended with a new context encoder, and the encoder for the source sentences is conditioned on this context encoder [3,4,8,9]. In addition, large-scale pretrained language models, such as BERT [10], have been used as context encoders for document-level machine translation models [6,7]. However, in an environment with more parameters than in BERT, these models have an underfitting problem because the optimal model cannot be obtained without transfer learning on the pretrained NMT model. To address these problems, in this study, a separate residual connection network called cardinality is used for each multihead attention module. Cardinality [11,12] refers to the number of network block groups and is a powerful operation for reducing the computational cost and number of parameters while maintaining a similar (or slightly better) performance. Thus, when training a model with a large number of parameters, the reduction in the error rate achieved is much better than that in other models with a deep and broad width. This notion leads to the understanding that cardinality has a positive effect on model learning of context representation when the number of parameters is increased with BERT. In addition, there are some problems with machine translation in the document-level context. NMT ignores the connections between sentences and other valuable contextual information. To address these problems, various

context-aware machine translation and document-level NMT models have been proposed to extract contextual information from neighboring sentences [13–15], focusing on using a few previous sentences as context for document-level machine translation. More recently, studies on document-level machine translation [6,7,16] have used multiple sentences as input data into a multiencoder. A commonly used representation of the input data is the concatenation of previous and current sentences. However, it cannot be confirmed with certainty that the two sentences are contextually close or are perfectly connected with the data. Ignoring contextual correlation between sentences can limit further improvements in translation performance. In a recent study, the index of similarity and unifying statistical theory of translation [17,18], based on communication theory, such as the signal-to-noise ratio (SNR), were proposed to assert mutual linguistic mathematical relationships. The statistical and linguistic features of a text depend not only on the particular language, mainly through a linear relationship, but also on the particular translation task. The communication channel used as the "sentences channel" [17,18] compares the number of sentences in any couple of texts for an equal number of words by considering the average relationships and their correlation. This motivates us to propose a simple and effective mathematic method to exploit document-level context. Similarity measures are used to capture sentences with highly relevant contextual information in the current sentence. This way, contextual document-level data are built to train the document-level NMT model.

This study demonstrates that an optimal embedding and similarity measurement can have a high translation performance for each data point, and the proposed model structure provides an advantage in document-level translation. The experimental results show that our simple and effective method improves the translation performance over the state-of-the-art baseline of the Transformer for IWSLT tasks and two benchmark machine translation tasks. In addition, an additional study was conducted to examine the ability of our method to capture document-level context information as well as to define the characteristics of the corpus.

The remainder of this paper is organized as follows. In Section 2, the background knowledge supporting the proposed approach and the existing document-level machine translation research is explained. In Section 3, the proposed method, which can improve the context-aware ability of document-level translation, is described. Finally, in Sections 4 and 5, details of the experiments and results are presented, respectively, followed by a conclusion in Section 6.

## 2. Related Work

### 2.1. Sentence Embedding

#### 2.1.1. Universal Sentence Encoder

The universal sentence encoder (USE) [19] model encodes textual data into high-dimensional vectors known as embeddings, which are numerical representations of the textual data. USE was trained on a variety of data sources to learn from a wide variety of tasks such as text classification, semantic similarity, and clustering with the sources consisting of Wikipedia, web news, web question–answer pages, and discussion forums. The input was a variable-length English text, and the output was a 512-dimensional sentence embedding. Because the embedding works on multiple generic tasks, it captures only the most informative features and discards noise.

#### 2.1.2. Sentence BERT

Sentence-BERT (SBERT) [20] is a modification of the BERT network that uses Siamese and triplet network structures to derive semantically meaningful sentence embeddings. SBERT is a twin network that allows two sentences to be processed simultaneously in the same manner. SBERT adds a pooling operation to the output of BERT and robustly optimized BERT (RoBERTa) to derive fixed-size sentence embeddings. The classification

objective function concatenates the sentence embeddings $u$ and $v$ with the element-wise difference $|u - v|$ and multiplies it with the trainable weight $W_t$:

$$o = \text{softmax}(W_t(u, v|u - v))) \tag{1}$$

*2.2. Similarity Measures*

2.2.1. Cosine Similarity

Cosine similarity measures the similarity between two nonzero vectors through the inner product space. Two vectors with the same orientation have a cosine similarity of 1, and two vectors at $90°$ have a similarity of 0. In contrast, the two vectors diametrically opposed to each other have a similarity of $-1$, independent of their magnitude. Cosine similarity is used particularly in a positive space, where the outcome is neatly bounded between 0 with 1. One of the reasons for the popularity of cosine similarity is that it is highly efficient in the evaluation of sparse vectors. The cosine similarity between $A$ and $B$ is defined as

$$\text{sim}(A, B) = \cos(\theta) = \frac{A \cdot B}{\| A \| \| B \|} \tag{2}$$

2.2.2. Euclidean Distance

In mathematics, the Euclidean distance between two points in Euclidean space is the length of a line segment. This can be calculated using the Pythagorean theorem, which is also known as simple distance. This is the best proximity measure when the data are dense or continuous. Formulas are used to compute the distance between different types of objects, such as the distance from a point to a line. The Euclidean distance between $A$ and $B$ is given by:

$$\text{dis}(A, B) = \sqrt{\sum_{i=1}^{k} (A_i - B_i)^2} \tag{3}$$

2.2.3. Manhattan Distance

The Manhattan distance is a metric in which the distance between two points is calculated as the sum of the absolute differences in their Cartesian coordinates. Simply stated, it is the total sum of the differences between the x and y coordinates. To calculate the Manhattan distance between two points $A$ and $B$, how these two points, $A$ and $B$, vary along the X- and Y-axes must be determined. Mathematically, the Manhattan distance between two points is measured along the axes at right angles and is represented as

$$\text{dis}(A, B) = \| A - B \|_1 = \sum_{i=1}^{n} |A_i - B_i| \tag{4}$$

2.2.4. Signal-to-Noise Ratio Distance

In statistical theory, the standard definition of signal-to-noise ratio (SNR) is the ratio of signal variance to noise variance, thus, the SNR between anchor feature $A$ and compared feature $B$ is

$$\text{SNR} = \frac{\text{Signal}}{\text{Noise}} \text{SNR}_{A,B} = \frac{\text{var}(A)}{\text{var}(A - B)}, \tag{5}$$

where $\text{var}(a) = \sum_{i=1}^{k} (a - \mu)^2 / n$ denotes the variance of $a$, and $\mu$ is the mean value of $a$.

The variance in information theory reflects informativeness. More explicitly, signal variance measures useful information, whereas noise variance measures useless information. Therefore, increasing the SNR can improve the ratio of useful information to useless information, which indicates that the compared feature can be more similar to the anchor feature. In contrast, decreasing the SNR can increase the proportion of noise information, leading to a greater difference between the two features. Therefore, the values of the SNR distance can be used to measure the difference in a pair of features reasonably, which is essential for constructing a distance metric.

Based on the definition of the SNR, the distance of the signal-to-noise ratio is given by $1/\text{SNR}$. The SNR distance features follow a zero-mean value and can be represented as

$$\text{dis}(A, B) = \frac{1}{\text{SNR}_{A,B}} = \frac{\sum_{i=1}^{k} (A_i - B_i)^2}{\sum_{i=1}^{k} (A_i)^2}. \tag{6}$$

### 2.3. Attention Layer of the Transformer

The Transformer consists of an encoder and decoder. The input of the encoder comprises source sentences, and the output of the encoder is the context matrix of the source language. The decoder takes the target tokens and the context matrix of the source language as input and provides the probability of the next word in the target language. Both the encoder and decoder are composed of multiple layers. For the encoder, every layer has a self-attention sublayer and position-wise feed-forward sublayer. For the decoder, every layer has a self-attention sublayer, an encoder–decoder attention sublayer, and a position-wise feed-forward sublayer. The self-attention and encoder–decoder attention sublayers have the same attention mechanisms. The attention mechanism is expressed as follows:

$$\text{Attention}(Q, K, V) = \text{softmax}\left(\frac{QK^T}{\sqrt{d_k}}\right) V. \tag{7}$$

Let Attention $(Q, K, V)$ describe the attention layer, where $Q$, $K$, and $V$ indicate the query, key, and value, respectively. The difference between self-attention and encoder–decoder attention is a parameter for calculating attention. In the self-attention sublayer of the encoder, $Q$, $K$, and $V$ are calculated by multiplying the input vector $X = [x_1, \ldots, x_n]$ by the weight matrices that are learned during training. In $[x_1, \ldots, x_n]$, $x$ is a token vector in the source sentence and $n$ is the length of the source sentence. Similar to an encoder, the self-attention of the decoder, $Q$, $K$, and $V$ are calculated by multiplying the input vector $Y = [y_1, \ldots, y_m]$ with the weight matrices that are learned during training. In $[y_1, \ldots, y_m]$, $y$ is the token vector of the target sentence, where $m$ is the length of the target sentence. In the encoder–decoder attention of the decoder, $Q$ is calculated from $[y_1, \ldots, y_m]$. $K$ and $V$ are calculated from $[x\prime_1, \ldots, x\prime_n]$, where $x\prime$ is the token vector of the last layer of the encoder. After calculating the attention sublayer, each layer has a residual connection followed by layer normalization.

### 2.4. Cardinality Residual Connection

Convolutional neural networks (CNN) [21] have emerged as the dominant algorithms in computer vision, and the development of recipes for their design has received considerable attention. By stacking convolution layers deeply, the power of the model can be strengthened, leading to performance improvement. However, stacking multiple layers causes problems such as excessive computation or increased learning instability. To minimize the error from the deeper layers during the learning process, the residual connection network (ResNet) [22] was proposed in which a skip connection is added to the existing CNN layer, such that in the resulting structure, the input is added to the stack of two convolutional networks. Although the depth of the CNN increases, it can solve the problem of the gradient loss that occurs during the learning process. This can be expressed as

$$H(x) = F(x) + x \tag{8}$$

ResNeXt [11] presents a simple structure that adopts the strategy of repeating layers [22] and uses the concept of cardinality, which refers to the size of a group of transformations. Even under the restriction of maintaining complexity, increasing cardinality can improve classification accuracy and is more effective than going deeper or wider when

the capacity is increased. The network is constructed by repeating a building block that aggregates a group of transformations. This can be expressed as (9).

$$H'(x) = \sum_{i=1}^{C} F_i(x) + x \tag{9}$$

Following (8), the residual unit obtains $F(x)$ by processing x using two weight layers; subsequently, $x$ is added to $F(x)$ to obtain $H(x)$. As $H(x) = F(x) + x$, obtaining the desired $H(x)$ depends on obtaining perfect $F(x)$.

The aggregated transformation in Equation (9) serves as a residual function. $F_i(x)$ can be an arbitrary function analogous to a simple neuron: $F_i$ projects $x$ onto an embedding and then transforms it, and $C$ is the number of cardinalities. A module in this network performs a set of transformations, each on a low-dimensional embedding, thereby aggregating the outputs by summation. This can compensate for the enhanced training speed under large-scale parameter settings. In training a model with a large parameter, the use of a cardinality residual connection has proven to be better than other models with a deep width.

*2.5. Document-Level Neural Machine Translation*

Developing a document-level NMT model is important in generating more consistent and coherent translations. To integrate document-level contextual information into the NMT model, a cache was used to selectively memorize the most relevant information in the document context [23,24]. In other studies, the previous sentence of the current sentence was used to extract the partial document context [3,5–7,25]. Single-encoder models [26–29] exploit the concatenation of multiple sentences as the NMT input data, whereas multien-coder models [8,13,16] integrate an additional encoder to leverage contextual information in NMT systems. Two different hierarchical attention models have also been used to encode an entire document using a selective attention network [4,5,15]. BERT was used to under-stand contextual representations by initializing the parameters of the document-level NMT model encoder [26,27,29]. By leveraging knowledge distillation, the knowledge acquired from BERT was transferred to NMT [30,31]. In [6], they exploited the representation from BERT, integrating it into the encoder and decoder of a Transformer model. Although these approaches have achieved some success in document-level machine translation, they suffer from incomplete document context. Moreover, the BERT-fused model has the underfitting problem, which occurred in training settings without transfer learning. The reason why original linguistic relationships have been lost and texts mathematically have been dis-torted in many languages and translations was found [14,15]. Because this theory addresses linear regression lines, the concepts of SNR [17,18,32] and likeness index [17] can be used to reasonably measure the difference in a pair of features. Based on this background, we benchmark the SNR method [17,18,32] and propose cosine similarity, Euclidean distance, and Manhattan distance, which are simple and effective mathematical methods, to exploit document-level context. Similarity measures are used to capture sentences with highly relevant contextual information in the current sentence. The proposed methods can help the translation model better understand document-level contextual representations.

## 3. Context Sentence and Context-Aware NMT Model

*3.1. Document-Level Sentences of Corpus by Similarity Measures*

Formally, let $X = \{x_1, x_2, \ldots, x_K\}$ denote the source document with $K$ sentences and $Y = \{y_1, y_2, \ldots, y_K\}$ denote the target document sentences, such that $(x_k, y_k)$ is assumed to be a parallel-sentence pair. Following [3], the target-side document-level context, $Y_{<k}$ can be omitted because the source-side document-level context $X_{<k}$ conveys the same information as $Y_{<k}$. To generate $y_k$, source document $X$ can be divided into three parts. First, the $k$-th source sentence is $X_{=k} = x_k$. Second, the source-side document-level context on the left is $X_{<k} = x_1, \ldots, x_{k-1}$. Finally, the source-side document-level context on the

right is $X_{>k} = x_{k+1}, \ldots, x_K$. Therefore, the document-level translation probability can be approximated as

$$P(Y|X) \approx \prod_{k=1}^{K} P(y_k|x_k; X_{<k}; X_{>k}) \tag{10}$$

As document-level contexts often include several sentences, it is important to capture long-range dependencies and identify relevant contextual information. In this study, a sentence similarity measure is proposed to identify sentences with highly relevant contextual information in the current sentence. As shown in Figure 1, all $K$ sentences in the document were converted into embedding vectors using a universal sentence encoder or Sent-BERT. The embedding vector is represented by $\hat{X} = \{\hat{x}_1, \ldots, \hat{x}_K\}$. The similarity score of the current sentence $x_k$ was then measured with respect to all the other sentences $(X_{<k}, X_{>k})$ in the document. The similarity score metrics consisted of cosine similarity, Euclidean distance, and Manhattan distance. The result of similarity $\check{x}_k$ in $k$-th source sentence is denoted by,

$$\begin{aligned} \widetilde{x}_{k,n} &= \text{sim}(\hat{x}_k, \hat{x}_n) \\ \check{x}_k &= \{\widetilde{x}_{k,1}, \ldots, \widetilde{x}_{k,n}, \ldots, \widetilde{x}_{k,N}\} \end{aligned} \tag{11}$$

where $\check{x}_k$ is a $1 \times K$ matrix of similarity scores; excepting the score obtained for the same sentence, the highest measured score was used to select the most similar sentence as the document-level context sentence $x_k^{context}$, according to

$$x_k^{context} = \max(\check{x}_k) \tag{12}$$

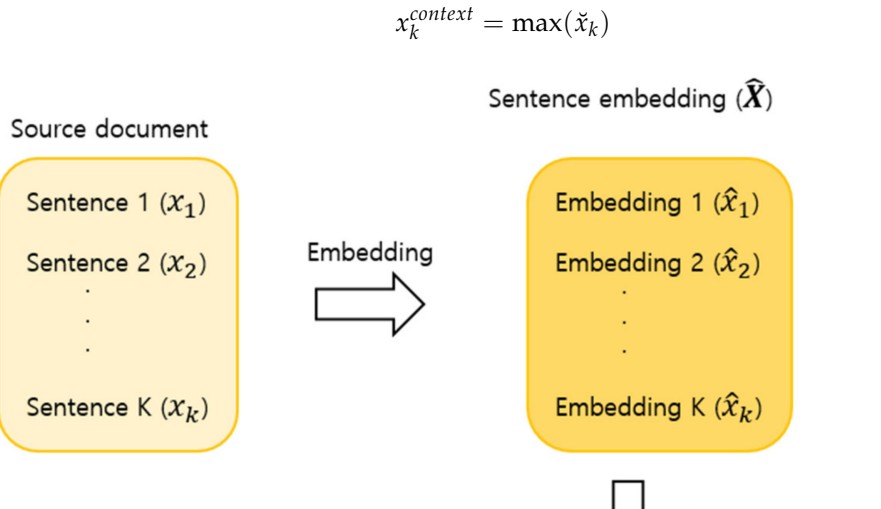

**Figure 1.** Process of finding the context sentence using the similarity score.

After the context sentences were selected for each sentence, both $x_k$ and its similar sentence $x_k^{context}$ were concatenated for the input format expressed as [CLS] $x_k$ [SEP] $x_k^{context}$ [SEP], where [CLS] and [SEP] are special tokens generated during the tokenization process of BERT.

### 3.2. Incorporating Contexts into NMT Model

In this section, we introduce the use of multiple document-level sentences as input for BERT and the incorporation of the context encoder with the NMT model. The structure of the model is shown in Figure 2.

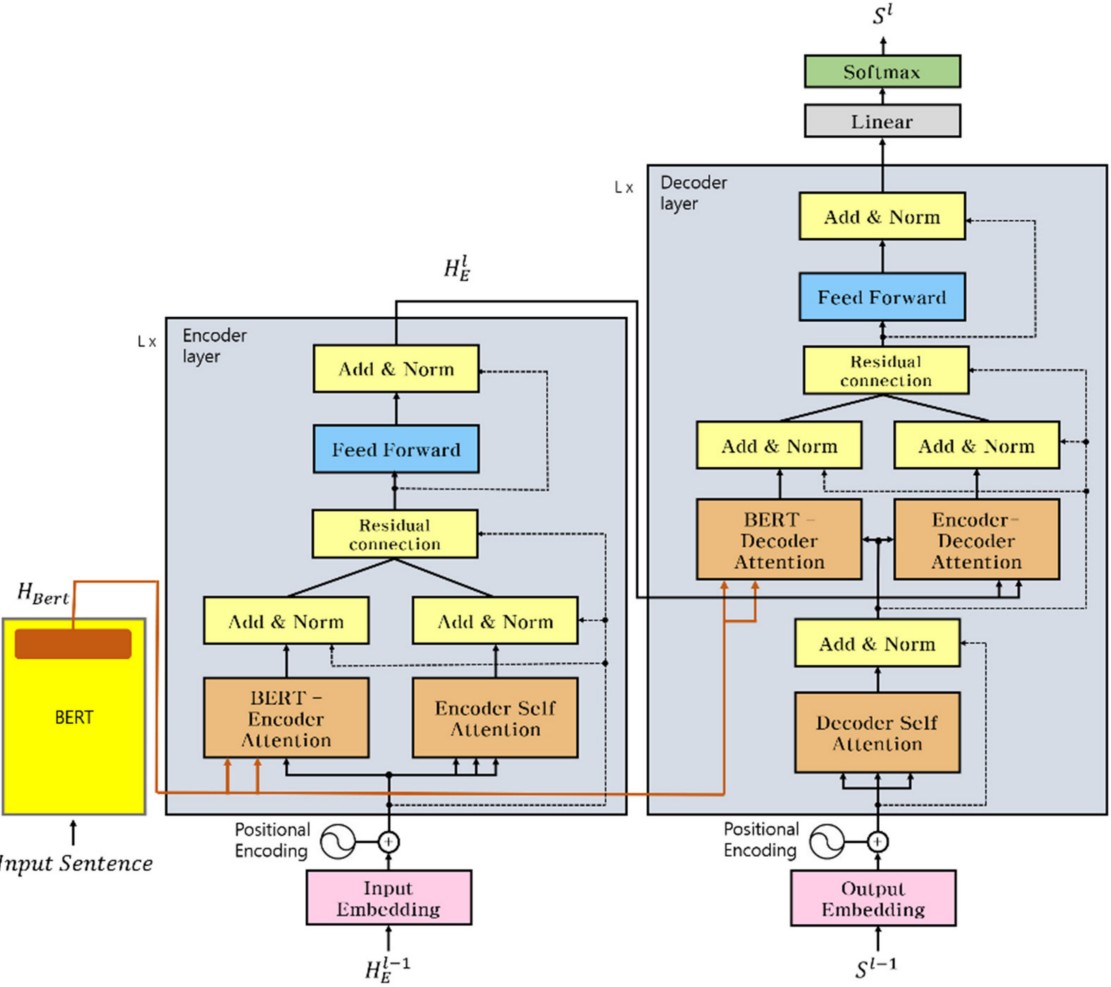

**Figure 2.** Incorporating a context encoder into the Transformer model by applying the cardinality residual connection.

The BERT context representation $H_B$ was integrated into both the encoder and decoder of the Transformer model using multiencoder approaches [3,6,8].

In the encoder, the first attention is on multihead self-attention. The second attention model is context attention, which incorporates BERT document-level context into the encoder; $Q$ is the input to the previous output of the hidden layer of the encoder; and $K$ and $V$ are the outputs of the last hidden layer of BERT, denoted by $H_B$. This can be represented as $attn_{B+E}\left(h_i^{l-1}, H_B, H_B\right)$. The two points of attention are as follows:

$$attn_E\left(h_i^{l-1}, H_E^{l-1}, H_E^{l-1}\right) \qquad (13)$$

$$attn_{B+E}\left(h_i^{l-1}, H_B, H_B\right) \qquad (14)$$

Similar to the encoder layer, the first attention of the decoder is the multihead encoder–decoder attention. The second attention model incorporates BERT into the decoder attention. $Q$ is the output of the hidden self-attention representation in the decoder. The key and values are the last hidden layers of BERT. This is denoted by $attn_{B+D}\left(s_t^{l-1}, H_B, H_B\right)$. These two types of attention are expressed as

$$attn_{E+D}\left(s_i^{l-1}, H_E^L, H_E^L\right) \tag{15}$$

$$attn_{B+D}\left(s_i^{l-1}, H_B, H_B\right) \tag{16}$$

The outputs of each attention are not combined immediately, but only after each attention passes through an additional normalization layer and residual connection. This is motivated by the cardinality concept, which is the structure used to calculate the residual connection separately. Let $\check{h}_i^l$ denote the residual connection from the perspective of a BERT encoder. Let $\hat{h}_i^l$ denote the residual connection of the self-encoder attention. Then $h_i^l$ is denoted by the $i$-th element in $H_E^l$, which is the hidden representation of the $l$-th layer in the encoder. Then, $\widetilde{h}_i^l$ is the result of summing both attentions in half. After computing the residual connection, the two results are summed in half. Thus, the output value $\widetilde{h}_i^l$ is obtained

$$
\begin{aligned}
\widetilde{h}_i^l &= \tfrac{1}{2}\left(\check{h}_i^l + \hat{h}_i^l\right) + h_i^{l-1} \\
\check{h}_i^l &= attn_{B+E}\left(h_i^{l-1}, H_B, H_B\right) + h_i^{l-1} \\
\hat{h}_i^l &= attn_E\left(h_i^{l-1}, H_E^{l-1}, H_E^{l-1}\right) + h_i^{l-1}
\end{aligned}
\tag{17}
$$

where $\widetilde{h}_i^l$ is inserted into the feed-forward network. Each attention was calculated using the residual connection and normalization. The encoder then outputs $H_E^l$ from the last layer. The decoder architecture is similar to that of the encoder. Let $s_t^l$ denote the hidden representation from self-attention in the decoder preceding time step $t$. This is the value obtained after positional encoding before self-attention. Positional encoding is embedded and added to the decoder inputs to indicate the position of each word. Let $\check{s}_i^l$ denote the residual connection from the BERT-decoder attention function.

$$
\begin{aligned}
\widetilde{s}_t^l &= \tfrac{1}{2}\left(\check{s}_t^l + \hat{s}_t^l\right) + s_t^l \\
\check{s}_t^l &= attn_{B+D}\left(s_t^l, H_B, H_B\right) + s_t^l \\
\hat{s}_t^l &= attn_{E+D}\left(s_t^l, H_E^l, H_E^l\right) + s_t^l
\end{aligned}
\tag{18}
$$

Let $\check{s}_i^l$ denote the residual connection from the encoder–decoder attention function, whereby $\widetilde{s}_t^l$ is input into the feed-forward to obtain $s_t^l$. Finally, $s_t^l$ passes through a linear layer and softmax layer to obtain the $t$-th predicted target word. The decoding process continues until the end of the sentence token. Because the parallel structure operation is performed on two attentions, the cardinality can be defined as Equation (9). As shown in Figure 3b,c, to minimize the loss resulting from the beneficial influence of incorporating context, a separate residual connection network [11] was used for each multihead attention of the encoder and decoder. Thus, more attention is paid to context representation. In addition, it is noted that the reduction in the error rate is much better than that in the other models, even with an increasing number of parameters.

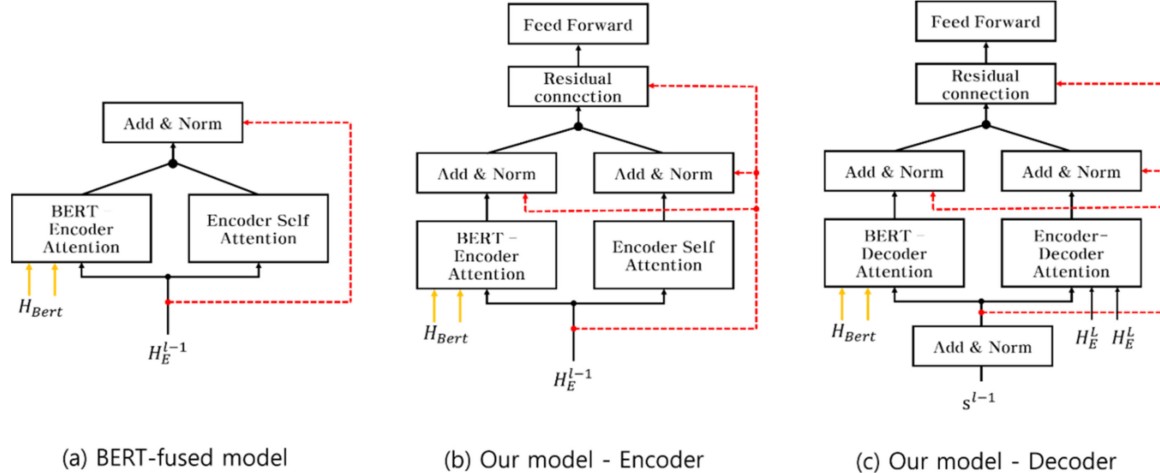

(a) BERT-fused model  (b) Our model - Encoder  (c) Our model - Decoder

**Figure 3.** Differences among residual connections. The residual connection is denoted by the red line. In (**a**), it do not use a separate residual connection. In (**b**,**c**), we use a separate residual connection structure for each attention layer of encoder and decoder.

## 4. Experiments

Document-level approaches were evaluated using several publicly available datasets. For English↔German (En↔De) and English↔Spanish (En↔Es), it was the IWSLT'14 (https://wit3.fbk.eu/home (accessed on 24 June 2022)) [33]. Evaluation campaigns were used as the training data. The data were partially extracted from the training set for validation. For the test set, dev2010, dev2012, and tst2010–2012 were concatenated in En↔De, and dev2010 and tst2010–2012 were concatenated in En↔Es. For English↔French (En↔Fr), IWSLT'17 [33] was used. The validation data and test set were merged from dev2010 and tst2010–2015. Following a previous study [4], the two English→German (En→De) benchmark datasets were used: TED and news commentary (https://opus.nlpl. eu/News-Commentary.php (accessed on 24 June 2022). The processed datasets were obtained from [4] so that the results could be compared with those of previous studies. For SNR distance, only 0.1 M sentences were used as a training set in the IWSLT'14 En→De model. The details of all datasets are listed in Table 1. Byte pair encoding [34] was used to segment all sentences and lowercase words. The merge operations for each dataset are listed in Table 2. The evaluation metric was BLEU [35].

**Table 1.** Statistics of document-level machine translation datasets.

| Type | Dataset | #Sent | Avg. #Sent |
|---|---|---|---|
| IWSLT'14 | En↔De | 0.16 M/7 K/6.7 K | - |
| | En→De | 0.1 M/7 K/6.7 K | - |
| | En↔Es | 0.17 M/8 K/5.5 K | - |
| IWSLT'17 | En↔Fr | 0.22 M/9.9 K/9 K | - |
| Maruf et al. (2019) | TED | 0.21 M/9 K/2.3 K | 121/96/99 |
| | News | 0.24 M/2 K/3 K | 39/27/19 |

**Table 2.** Merge operation of each dataset.

| Type | Dataset | Merge Operation |
|---|---|---|
| IWSLT'14 | En↔De | 10,000 |
| | En↔Es | 10,000 |
| IWSLT'17 | En↔Fr | 10,000 |
| Maruf et al. (2019) | TED | 30,000 |
| | News | |

The design of the BERT model was specific to the source language. BERT-base-uncased was used for the model in which the source language was English, and BERT-base-multilingual-uncased was used for Spanish and French as source languages. The models were created by Google. We used the BERT-base-german-cased model [36] in IWSLT'14 De-En, where the source language is German. BERT was trained using the latest German Wikipedia dump (6 GB), OpenLegalData dump (2.4 GB), and news articles (3.6 GB). It was created by Deepset, which is the company behind the Haystack NLP framework. For our model (https://github.com/ChoiGH/CATSBY (accessed on 24 June 2022)), we used the Fairseq [37] implementation of transformer network. All experiments are run on a dual NVIDIA TITAN X GPU with 24 GBs of memory. It takes about 18 h to train the IWSLT'14 En→De model. The batch size was 4096 tokens per GPU and model configuration is the same as was used in [4]. The batch size was 4000 tokens. The hidden size was 512, and the feed-forward network layer dimension was l024. The embedding size was 512; the number of attention heads was four; and the dropout rate [38] was 0.3. The number of layers for the encoder and decoder was six, and the Adam [39] optimizer was used with momentums $\beta_1 = 0.9$ and $\beta_2 = 0.98$. The same learning rate schedule strategy as in [2] was applied with 4000 warmup steps for label smoothing of the cross-entropy loss with a smoothing rate of 0.1.

## 5. Results and Discussion

In this paper, we propose a new BERT-Transformer model using the distributed residual connection structure inspired by the concept of cardinality and a method for defining contextual data through sentence embedding and similarity measurement. The proposed model structure is very different from previous study [6] in that it does not transfer-learning pretrained NMT. The model [6] without transfer learning has an underfitting problem, but the proposed model with distributed residual connection can reflect contextual features to model without the underfitting problem. In addition to the advantages of the proposed model structure, the optimal data structure for document-level translation task is also proposed. The input of BERT is used not only as a single sentence but also as a dual sentence. To make dual sentence form, previous studies define adjacent sentences of sentence as context, but they cannot be sure that sentences are perfectly related [3,5–7]. In order to solve this problem, we propose a mathematical method to utilize document-level contextual information. The similarity measurement is used to detect sentences containing high contextual relevance in current sentence. Through the experiment, we prove the superiority of the proposed model compared with previous studies. In particular, it proves that the similarity measurement reflects the contextual relevance of data better than other methods. Finally, when defining the context sentence according to the range of a document, we present a new discovery that it can be different for each characteristic of the data.

### 5.1. The Reason for Using Similarity Measurment

SNR theory can be explained by supporting the use of our similarity measurement method. This motivated us to use three similarity measurement methods. In our method using sentence embedding, the SNR distance [32] is used to find the context sentences. However, the SNR distance requires considerable computing power; therefore, we had to reduce the size of the corpus and use only USE sentence embedding. The IWSLT'14

English–German corpus was used to prove the effectiveness of the similarity measurement method. Table 3 shows that the similarity measurement methods are superior to that of previous studies. We compare the method of *Context* [3,4,6,7], which is represented by the surrounding sentences of the source sentence. The BLEU score of the SNR distance is higher than that of *Context*. Based on this result, we propose three similarity measurement methods as a manner of building a context-aware document-level corpus for a training model.

**Table 3.** BLEU scores of IWSLT'14 English to German model by type of context.

| Type of Context | Context | Similarity | | | |
|---|---|---|---|---|---|
| **Sentence Embedding** | **-** | **Universal Sentence Encoder** | | | |
| Similarity Measurement Method | - | SNR | Cosine | Euclidean | Manhattan |
| BLEU score | 27.95 | 28.19 | 28.64 | 28.58 | 28.4 |

*5.2. Best Combination for Document-Level Data*

In this section, an embedding method and similarity measurement are defined to generate optimal context-aware data for each model. As shown in Figure 4, a heatmap graph matrix can be created by measuring the similarity between the sentences. IWSLT'14 En→Es is used as an example. In this state, BERT embedding and cosine similarity are applied. The matrix indicates the degree of similarity between sentences. Based on the similarity score, the sentence with the higher similarity scores, except one, was chosen as the context sentence. The similarity within the range of each training, validation, and test dataset was measured.

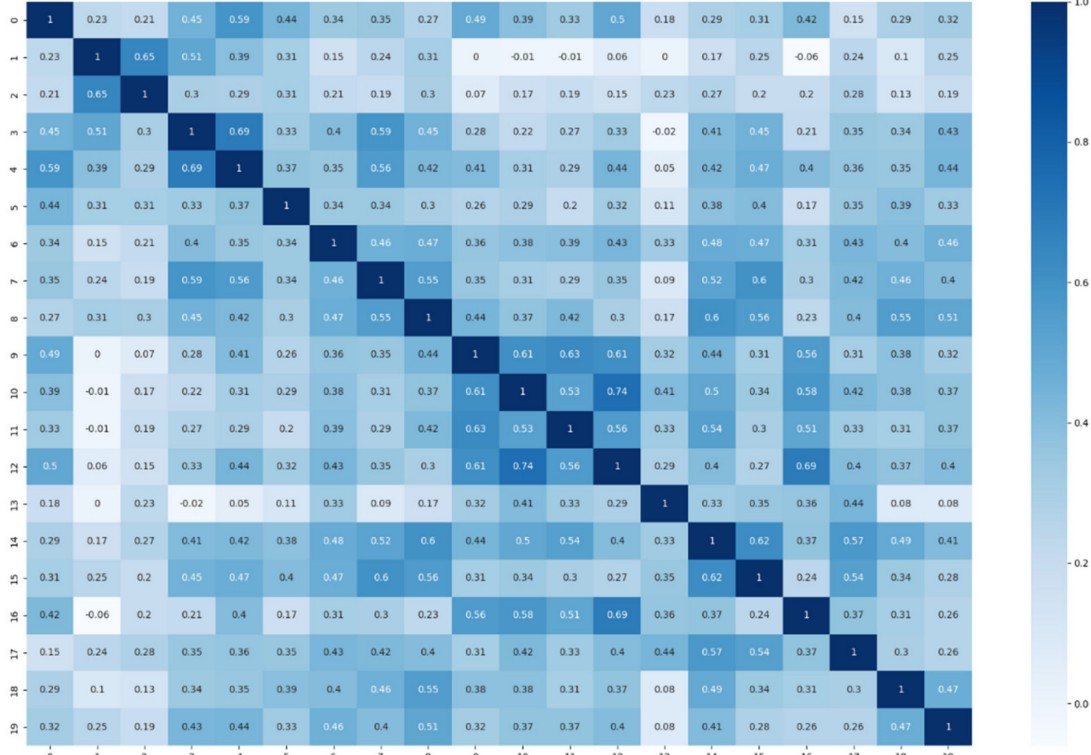

**Figure 4.** Heatmap matrix of the scores assigned to each sentence after measuring the similarity. Only 20 sentences were extracted.

Table 4 shows the BLEU score of the model trained on data to which two embedding methods and three similarity measurement methods were applied. For En↔De, the use of a universal sentence encoder and cosine similarity outperformed the other methods. Sentence-BERT and cosine similarity outperformed the other methods in En↔Es and En→Fr. In Fr→En, Sentence-BERT and the Euclidean measure outperformed other methods. Methods optimized for the translation model could be determined. In general, the cosine similarity measure scored a high BLEU for most models. Based on these results, the embedding and similarity measure method with the highest score was used for each translation model to build the data for the next experiment.

**Table 4.** BLEU scores of the models. Best embedding and similarity measure for each model. "Cos", "Euc", and "Manh" indicate the cosine similarity method, Euclidean distance, and Manhattan distance, respectively.

| Model | Transformer (Vaswani et al., 2017) | USE | | | BERT | | |
|---|---|---|---|---|---|---|---|
| | | Cos | Euc | Manh | Cos | Euc | Manh |
| IWSLT'14 En→De | 28.59 | 30.85 | 29.99 | 30.09 | 30.29 | 30.2 | 30.05 |
| IWSLT'14 En→Es | 37.36 | 39.59 | 39.17 | 39.36 | 39.67 | 39.23 | 39.06 |
| IWSLT'17 En→Fr | 40.54 | 43.68 | 43.51 | 43.49 | 43.91 | 43.4 | 43.26 |
| IWSLT'14 De→En | 34.4 | 35.92 | 35.2 | 34.9 | 35.69 | 35.81 | 35.56 |
| IWSLT'14 Es→En | 40.99 | 40.66 | 40.2 | 40.08 | 42.14 | 41.73 | 41.95 |
| IWSLT'17 Fr→En | 41.00 | 42.09 | 41.55 | 41.46 | 42.03 | 42.36 | 42.09 |

*5.3. Effect of the Proposed Methods on Document-Level Translation*

The results of the experiments compare seven context-aware NMT models consisting of the document-aware Transformer [3], selective attention NMT [4], hierarchical attention NMT [5], query-guided capsule network [31], flat-Transformer [27], the flat-Transformer initialized using the BERT encoder, BERT-fused [6], and BERT-fused applied to the context gate [7]. Most of the results of previous studies are from [7]. The proposed model was compared with the aforementioned baseline methods. Table 5 summarizes the results of these models. The results show that our model obtained BLEU scores of 27.23/27.98 on the two datasets and significantly outperformed the other models listed in the table, achieving state-of-the-art performance.

**Table 5.** BLEU scores of two document-level machine translation benchmarks.

| Model | TED | News |
|---|---|---|
| RNN (Bahdanau et al., 2015) | 19.24 | 16.51 |
| HAN (Werlen et al., 2018) | 24.58 | 25.03 |
| SAN (Maruf et al., 2019) | 24.62 | 24.84 |
| QCN (Yang et al., 2019) | 25.19 | 22.37 |
| Transformer (Vaswani et al.,2017) | 23.28 | 22.78 |
| Flat-transformer (Ma et al., 2020) | 24.87 | 23.55 |
| + BERT (Ma et al., 2020) | 26.61 | 24.52 |
| BERT-fused model (Zhu et al., 2020) | 25.59 | 25.05 |
| + Context gate (Zhiyu et al., 2021) | 26.23 | 26.55 |
| Our model | 27.23 | 27.98 |

*5.4. Capturing the Contextual Information*

To investigate whether our method can capture contextual information in the BERT context encoder, the experimental settings in [16], referred to as *Context*, were adopted. This implies the concatenation of previous and current sentences. In addition, two types

of contexts were presented as inputs for the context encoder. One is *Similarity*, which is our methodology, and the other is *Same*, which concatenates the current sentence to the current sentence. This experiment is to check whether the main cause is to improve the quality of data by our proposed method rather than simply to increase the amount of information in the data. Three types of inputs were provided to the BERT context encoder, and experiments were conducted on the IWLST'14 and IWSLT'17 datasets. As shown in Table 6, the performance of our method was better than those of the other methods. A corpus that does not consider context does not achieve significant performance improvement over the proposed method. Although the previous sentence is sometimes contextually similar to the current sentence, this situation is not always guaranteed. Furthermore, the use of contextually incorrect sentences results in a significant performance penalty.

**Table 6.** BLEU scores of models with three types of context.

| Model | Type of Context | | |
| (IWSLT) | Similarity | Context | Same |
|---|---|---|---|
| En→De | 30.85 | 30.12 | 29.45 |
| En→Es | 39.67 | 37.09 | 37.21 |
| En→Fr | 43.91 | 41.32 | 41.37 |
| De→En | 35.92 | 35.13 | 35.13 |
| Es→En | 42.14 | 41.83 | 41.46 |
| Fr→En | 42.36 | 41.69 | 41.6 |

*5.5. What If the Data Created by Measuring the Similarity within Diverse Talk Ranges in Data Are Used?*

In the above experiments, a similarity measure was used for the entire corpus. There are several TED talks within the corpus, as listed in Listing 1, where the original file contains the talk information provided between the tags. We used this feature to select context sentences, experimenting with two methods to choose context sentences by measuring their similarity within diverse talk ranges. First, the talks in the IWSLT corpus were separated, thereby extracting context sentences within the range of each talk and checking whether the document-level training data would change the translation performance. It was assumed that if sentences in the same domain were combined, they would have the potential to be the features of a document. After extracting the data using talkid tags, context sentences can be chosen using similarity measurements. Table 7 shows the differences in context sentences extracted by the range of talk. A point to verify is that there are cases in which the results are the same as those calculated using the similarity in all talks. Clearly, these sentences are very similar in all cases. Second, it is difficult to find a contextual sentence near the beginning of the talk within the range of the lecture. An example is a greeting or a background explanation before starting the main story. Therefore, a hybrid method is proposed to extract contextual sentences at the beginning of a conversation. Table 8 shows an example of extracting the context sentences from the four sentences previously uttered at the beginning of the entire talk. It is observed that the contextual sentences extracted from the entire talk are closer in context than those extracted from the range of one talk. These data were used to train the model. The performance of the models was then checked with respect to the context sentence extraction range. The IWSLT'14 En→De, En→Es, and IWSLT'17 En→Fr were used in the experiments. In the second experiment, context sentences were extracted differently from the beginning of the talk to the 2nd to 10th sentences. Table 9 shows the BLEU scores of the models trained with the data extracted using the context sentence selection methods used thus far. The values in the column with "All talks" are the results of Section 5.2. Surprisingly, the methods that obtained the highest BLEU scores differed among the models. In the English-to-German model, a high BLEU score was recorded when a context sentence was defined over the entire data range. In the English-to-Spanish model, a high BLEU score was recorded when the six sentences from the beginning of the talk were defined in the entire data range as contextual sentences. In the

English-to-French model, a high BELU score was recorded when a contextual sentence was defined for each talk. Thus, it can be proven that even if a context sentence is defined in the same domain, it does not always provide a good effect for all corpora. We were also able to understand the characteristics of this corpus. The English–German corpus has contextually similar sentences that are evenly distributed over the entire range of the document. It can be inferred that the English–Spanish corpus tends to have a weak relationship with the main content of the talk at the beginning of each talk. Finally, the English–French corpus tends to have a strong contextual relationship in each talk. This experiment confirmed that there is an appropriate method for defining context sentences based on the characteristics of the corpus.

**Table 7.** Differences in the context sentence extracted within the range of the talk.

| Sentence | Context Sentence | |
|---|---|---|
| | **Extracted in All Talks** | **Extracted in One Talk** |
| it can be a very complicated thing, the ocean. | and it can be a very complicated thing, what human health is. | and it can be a very complicated thing, what human health is. |
| and it can be a very complicated thing, what human health is. | health studies from the region are conflicting and fraught. | it can be a very complicated thing, the ocean. |
| and bringing those two together might seem a very daunting task, but what i'm going to try to say is that even in that complexity, there's some simple themes that i think, if we understand, we can really move forward. | well, right, it is a good thing to do, but you have to think what else you could do with the resources. | but in fact, if you look around the world, not only are there hope spots for where we may be able to fix problems, there have been places where problems have been fixed, where people have come to grips with these issues and begun to turn them around. |
| and those simple themes aren't really themes about the complex science of what's going on, but things that we all pretty well know. | and the answer is not complicated but it's one which i don't want to go through here, other than to say that the communication systems for doing this are really pretty well understood. | that's a good thing for this particular acute problem, but it does nothing to solve the pyramid problem. |
| and i'm going to start with this one: if momma ain't happy, ain't nobody happy. | now, if your mother ever mentioned that life is not fair, this is the kind of thing she was talking about. | and if we just take that and we build from there, then we can go to the next step, which is that if the ocean ain't happy, ain't nobody happy. |

**Table 8.** Examples of four sentences extracted using the hybrid method.

| Sentence | Context Sentence | |
|---|---|---|
| | **Hybrid Method** | **Extracted in One Talk** |
| it can be a very complicated thing, the ocean. | and it can be a very complicated thing, what human health is. | and it can be a very complicated thing, what human health is. |
| and it can be a very complicated thing, what human health is. | health studies from the region are conflicting and fraught. | it can be a very complicated thing, the ocean. |

**Table 8.** *Cont.*

| Sentence | Context Sentence | |
|---|---|---|
| | **Hybrid Method** | **Extracted in One Talk** |
| and bringing those two together might seem a very daunting task, but what i'm going to try to say is that even in that complexity, there's some simple themes that i think, if we understand, we can really move forward. | well, right, it is a good thing to do, but you have to think what else you could do with the resources. | but in fact, if you look around the world, not only are there hope spots for where we may be able to fix problems, there have been places where problems have been fixed, where people have come to grips with these issues and begun to turn them around. |
| and those simple themes aren't really themes about the complex science of what's going on, but things that we all pretty well know. | and the answer is not complicated but it's one which i don't want to go through here, other than to say that the communication systems for doing this are really pretty well understood. | that's a good thing for this particular acute problem, but it does nothing to solve the pyramid problem. |
| and i'm going to start with this one: if momma ain't happy, ain't nobody happy. | and if we just take that and we build from there, then we can go to the next step, which is that if the ocean ain't happy, ain't nobody happy. | and if we just take that and we build from there, then we can go to the next step, which is that if the ocean ain't happy, ain't nobody happy. |

**Table 9.** BLEU scores of methods that define the context sentence within a limited range of talks.

| Model | Talk Range for Similarity Measurement | | | | | | | | | | | |
|---|---|---|---|---|---|---|---|---|---|---|---|---|
| | **All Talks** | **Each Talk** | **Hybrid The $N$th Sentence from the Front)** | | | | | | | | | |
| | | | $N = 2$ | 3 | 4 | 5 | 6 | 7 | 8 | 9 | 10 |
| IWSLT'14 En→De | 30.85 | 30.29 | 30.49 | 30.44 | 30.44 | 30.53 | 30.48 | 30.75 | 30.41 | 30.55 | 30.52 |
| IWSLT'14 En→Es | 39.67 | 39.55 | 39.72 | 39.71 | 39.65 | 39.57 | 39.83 | 39.79 | 39.7 | 39.73 | 39.52 |
| IWSLT'17 En→Fr | 43.91 | 44.05 | 43.55 | 43.88 | 43.67 | 43.63 | 43.69 | 43.58 | 43.82 | 43.76 | 43.73 |

*5.6. View of the Similarity Measure*

When building a document-level corpus using similarity measures, we did not apply it to a large-scale corpus. To calculate the similarity score between the number of $n$ sentences, it is necessary to create an $n \times n$ matrix, which requires considerable computing power in memory. The largest dataset used in our experiment was approximately 250 K sentences merged with training and validation. An excess of this leads to out-of-memory problems. Therefore, we believe that sentence-similarity measures are appropriate for low-resource language pairs. Measuring similarity using a large-scale corpus is the next challenge.

**Listing 1.** Appearance of the original file. There are many talks separated by tags.

```
<doc docid="535" genre="lectures">
<description>TED Talk Subtitles and Transcript: At TED2009, Al Gore presents updated slides
from around the globe to make the case that worrying climate trends are even worse than
scientists predicted, and to make clear his stance on "clean coal."</description>
<talkid>535</talkid>
<title>Al Gore: What comes after An Inconvenient Truth?</title>
<reviewer></reviewer>
<translator></translator>
<seg id="1"> Last year I showed these two slides so that demonstrate that the arctic ice cap,
which for most of the last three million years has been the size of the lower 48 states, has shrunk
by 40 percent. </seg>
<seg id="2"> But this understates the seriousness of this particular problem because it doesn't
show the thickness of the ice. </seg>
<seg id="3"> The arctic ice cap is, in a sense, the beating heart of the global climate system.
</seg>
. . .
<seg id="90"> If you want to go far, go together." </seg>
<seg id="91"> We need to go far, quickly. </seg>
<seg id="92"> Thank you very much. </seg>
</doc>
<doc docid= "531" genre="lectures">
<description>TED Talk Subtitles and Transcript: In this short talk from TED U 2009, Brian Cox
shares what's new with the CERN supercollider. He covers the repairs now underway and what
the future holds for the largest science experiment ever attempted.</description>
<talkid>531</talkid>
<title>Brian Cox: What went wrong at the LHC</title>
<reviewer></reviewer>
<translator></translator>
<seg id="1"> Last year at TED I gave an introduction to the LHC. </seg>
<seg id="2"> And I promised to come back and give you an update on how that machine worked.
</seg>
<seg id="3"> So this is it. And for those of you that weren't there, the LHC is the largest scientific
experiment ever attempted – 27 km in circumference. </seg>
. . .
```

## 6. Conclusions, Limitations, and Future Research

In this study, we proposed a distributed residual connection that can utilize the contextual understanding ability of BERT more effectively in the NMT model. It was demonstrated that the performance of context-specific language model can be improved when the language comprehension ability of each attention is strengthened by applying independent residual connection to the two attention layers. In addition, a context data augmentation using similarity measurement method was proposed to obtain a translation result reflecting the context information of the document domain in the document-level translation. It was confirmed through experiments that the model learned using the corpus constructed in this way showed improved results compared with the existing research results. Finally, we can prove that the data augmentation method that defines the context sentence using the distributed residual connection method and the similarity measurement method is the optimal method to implement the NMT model that understands the context in translation tasks. In particular, experiments on two benchmark corpora demonstrated that our proposed method can significantly improve document-level translation performance compared with several document-level NMT baselines. In addition, the characteristics of the corpus were confirmed using a model trained on a corpus created by setting different ranges for the similarity measure.

Although contextual sentences defined using similarity measure and contextual understanding of BERT were improving the NMT model, the supplemented contextual information is not fully reflected in the decoder. Because the context features generated

by BERT are reflected only on the exposed training data in the NMT training process, the context to be reflected in the sentence cannot be created as intended.

To solve our limitations, we will try to improve the ability of the decoder's language generation by self-supervised learning using a target monolingual corpus. As a way to help the decoder understand the context feature, combining the GPT language model based on the transformer's decoder should be considered. Through this, not only the understanding of the context in the encoder but also the research on generating the translation word reflecting the context phenomena in the decoder should be conducted. Moreover, instead of the similarity method, a method of extracting context features based on the data of the vocabulary and phrase unit suitable for the document-level can be proposed. It is necessary to develop a model that is dependent on the domain area of the document and can be extended. Additionally, the problem of incorrect translation generation that can be caused by knowledge that is not covered by the domain should be continuously considered.

**Author Contributions:** Conceptualization, Y.-K.K., J.-H.S. and G.-H.C.; methodology, J.-H.S. and G.-H.C.; software, G.-H.C.; validation, Y.-K.K., J.-H.S. and G.-H.C.; formal analysis, Y.-H.L., J.-H.S. and G.-H.C.; investigation, G.-H.C.; resources, G.-H.C.; data curation, J.-H.S. and G.-H.C.; writing—original draft preparation, Y.-K.K., J.-H.S., Y.-H.L. and G.-H.C.; writing—review and editing, Y.-K.K., J.-H.S., Y.-H.L. and G.-H.C.; visualization, G.-H.C.; supervision, Y.-K.K.; project administration, Y.-K.K.; funding acquisition, Y.-K.K. All authors have read and agreed to the published version of the manuscript.

**Funding:** This work was supported by Electronics and Telecommunications Research Institute (ETRI) grant funded by the Korean government. [22ZS1140, Core Technology Research for Self-Improving Integrated Artificial Intelligence System].

**Conflicts of Interest:** The authors declare no conflict of interest.

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
