# Peer review of "Toward Understanding Most of the Context in Document-Level Neural Machine Translation"

_electronics, doi:10.3390/electronics11152390_

Round 1

Reviewer 1 Report

The paper is very well written, and proposes a simple and effective method for capturing contextual relationships to the field of neural machine translation. And the proposed solution has been experimentally demonstrated to outperform the prior art and to significantly improve the performance of machine translation. The paper can be accepted after minor revision.

1. The abstract can be more concise.

2. In the introduction, “To address these problems, various 48 document-level NMT models have been proposed to extract contextual information from 49 neighboring sentences [3–5,11,12,37]”. Bulky citations are not informative. Please separately explain the results of a single document for the convenience of readers.

3. The formatting needs to be corrected and/or improved.

Author Response

Response to Reviewer 1 Comments

----------------------------------------------------------------------------------------We appreciate the constructive comments which have contributed significantly to improving our paper. The replies to the comments are as follows:

----------------------------------------------------------------------------------------

Point 1 : The abstract can be more concise.

Response 1 : We made the abstract concise.

Point 2 :  In the introduction, “To address these problems, various document-level NMT models have been proposed to extract contextual information from neighboring sentences [3–5,11,12,37]”. Bulky citations are not informative. Please separately explain the results of a single document for the convenience of readers.

 Response 2 : We separated references for the convenience of readers.

Point 3 :  The formatting needs to be corrected and/or improved.

 Response 3 : We made several modifications to fit the format of the journal.

Reviewer 2 Report

Report on the manuscript "electronics-1808959" entitled "Toward Understanding Most of the Context in Document-Level Neural Machine Translation"

This manuscript introduces a method to capture the contextual correlation of a sentence to learn an effective contextual representation. A model for a separate residual connection network is proposed to minimize the influence of embedding context. The experimental outcomes indicate the good performance of the proposal. Conclusions about the present investigation are reported.

In general, I have a good opinion about this work. However, before its acceptance, the following issues must be fixed:

1. The manuscript needs to be proofread by the authors carefully. I have noted many drafting problems. I suggest the authors ask for assistance from a native English speaker.

2. Words in the title are not usually in the keywords. In addition, the keywords are often written in alphabetical order.

3. The authors must check the use of all acronyms, abbreviations, and notations employed in the whole manuscript. 

4. Texts used in formulas must be written in text style and not in math style (see, just as an example, "softmax" in Eq. (1); "sim" and "cos" in Eq. (1), etc.). Several of these issues are present in the whole manuscript and they must be fixed.

5. Fractions used in a line of text must be written with "/" instead of using a fraction; see, for example, lines 128 and 138.

6. I see the sections are somewhat disproportionate and unorganized. For example, some sections have subsections and others do not, even some of them have subsubsections. Some sections are too short. I suggest removing subsubsections and merging some sections. This decreases the quality of the presentation of the manuscript. I recommend the authors to look for a manner for organizing their manuscript so that its readability and presentation are improved.

7. To the best of my knowledge, there has been a good body of work done in the literature on this topic. The novelty and contribution of this study must be clearly stated. The authors must establish what is novel in their investigation from the theoretical, methodological, and applied points of view. All of them or only in the application? 

8. The authors must provide more details about the computational framework used in the manuscript. For example, software and packages used, features of the computer employed, runtimes, and other computational aspects must be added. 

9. With the aim of reproducibility, the authors should provide access to the data set and to the computational codes used.

10. I do not have checked each numerical result in detail. I recommend the authors to check them.

11. In my opinion, the implications of the study are underdeveloped and must be improved and explained further in the final discussion.

12. The conclusions need to be improved. Also, the authors must add more limitations to the study and more ideas for further research. Then, I suggest titling the final section "Conclusions, limitations, and future research".

13. The authors must check whether all references are cited and whether all citations are in the reference list.

Author Response

We appreciate the constructive comments which have contributed significantly to improving our paper. 

Reviewer 3 Report

The article proposes a new method to capture the contextual correlation of the sentence during machine translation of text documents.

The purpose of the study is presented in the introduction

The second and third points need to be combined, and the resulting new point can be called "Related Work". It would be good to shorten the material from point 2, because there are known concepts in it and the material has the characteristics of a textbook.

It would be good to expand the title of the point "Approach" and say what this approach is about.

Figures 3, 4, as well as some of the tables are wider than the text and go out of the print area which should be fixed.

In point 5, there is no need to have separate sub-points, because sub-points 5.1, 5.2, 5.3 are small in volume.

In Tables 7 and 8, there is no column spacing and the text merges.

In the conclusion it is said that the proposed method is "simple and effective"; it's unclear whether this is the case for all text documents. Usually in text processing there are a lot of specifics, so it would be better to say that for the experiments made the method gives good results

Author Response

(The authors gave the same response as above.)
